# Curative treatment incorporating subjective decisions on age and frailty is not beneficial for older patients with oral cavity squamous cell carcinoma

Alice Prevost[1]*, Hugo Poncet[1], Victor Benvegnu[1], Vinciane Poulet[1], Jacqueline Butterworth[1], Andréa Varazzani[2], Frédéric Lauwers[1], Franck Delanoë[1]

1 Plastic and Maxillofacial Surgery Department, Pierre-Paul Riquet Hospital, Toulouse University Hospital, Toulouse, France, 2 Plastic and Maxillofacial Surgery Department, Lyon Sud Hospital, Hospices Civils de Lyon, Claude-Bernard Lyon 1 University, Pierre-Bénite, France

* prevost.a@chu-toulouse.fr

## Abstract

The curative surgical treatment of older patients with oral cavity squamous cell carcinoma (OCSCC) is often personalized by incorporating subjective decisions on age and frailty. We aimed to determine here whether real-world recommended treatment, following official French guidelines only, versus deviation from recommended treatment was beneficial for older patients with OCSCC. To do this, we performed a retrospective evaluation of patients >70 years managed for treatment of p16-negative OCSCC in our tertiary hospital center in France between 2007 and 2017. The association between postoperative morbidity and deviation from recommended treatment was analysed using multivariate logistic regression. Cox Proportional Hazards Regression assessed the associations between deviation from recommended treatment and both the hazard of recurrence and mortality within 5 years. We included 185 patients who were recommended surgical resection of OCSCC: n = 147/185 (79%) patients underwent the recommended treatment and 38/185 (21%) patients underwent deviation from recommended treatment. Patients who underwent deviation from recommended treatment had a significantly lower recurrence-free survival (p = 0.0005) and overall survival (p = 0.008). Deviation from recommended treatment was found independently associated with increased development of 3-month postoperative morbidity (adjusted odds ratio 2.63 [1.23–5.82]; p = 0.02) and increased risk of recurrence within 5 years (adjusted hazard ratio 1.79 [1.14–2.83]; p = 0.01). Deviation from recommended treatment was not found independently associated with increased risk of mortality within 5 years (1.35 [0.82–2.23]; p = 0.2). Overall, deviation from recommended treatment was associated with worse outcomes and so we have identified a decision-making process biased by undocumented and subjective evidence. Preoperative risk models therefore require further validation in older patients with OCSCC to define more appropriate treatment regimens.

**Data availability statement:** All relevant data are within the paper and its Supporting information files.

**Funding:** The author(s) received no specific funding for this work.

**Competing interests:** The authors have declared that no competing interests exist.

## Introduction

Demographic projections suggest that new cancer cases among the elderly will account for almost 60% of global cancer incidence by 2035 [1] The number of annual head and neck cancer diagnoses is also expected to double by 2040 [2]. Currently, approximately 30% of patients with head and neck squamous cell carcinoma (HNSCC) are aged ≥70 years [1]. This demographic expansion raises practical, economic, and medical issues, as well as ethical questions related to therapeutic objectives and quality of life [3].

A major challenge in geriatric oncology is assessing the benefit/risk balance of indicated treatments for individual patients. Ageing is a highly heterogeneous process and therapeutic decisions must take into account individualized ageing, physiological reserves, and the potential treatment risks. The International Society of Geriatric Oncology regularly updates their official treatment guidelines in light of building a global consensus that is based on scientific evidence. However, the authors stress a lack of data for head and neck cancers [4] This is given the lack of randomized data for this sub-population, mainly due to the under-representation of older patients in clinical trials. Indeed, patients aged ≥70 years still represent <5% of participants in HNSCC clinical trials, as shown by the MACH-NC meta-analysis [5]. Thus, data extrapolated from HNSCC clinical trials may lack validation in older patients.

Despite consensus on the multimodal approach to the treatment of HNSCC (combining surgery ± radiotherapy ± chemotherapy), there remains uncertainty about the optimal treatment regimen for older patients. This frequently leads to under-treatment of the elderly [6], which can be partly explained by basing treatment choice on chronological age instead of medical assessments and patient preferences [7]. Older patients are associated with frailty and deteriorated autonomy. This age discrimination leads to prejudices on what treatment is most beneficial for older patients and treatment tolerances [8]. Indeed, less risky and less aggressive interventions may appear more reasonable options from the physician's viewpoint [9]. Accordingly, studies in digestive and gynecological oncology have shown that the indication of surgical treatment options decreases with age [10] and that the rate of curative surgery is significantly reduced in older patients [11]. Adjuvant treatments have also been found less considered and used [12].

These same therapeutic attitudes have been observed in the management of HNSCC in older patients [13–15]. Including 54,741 patients for study, Barrett et al. [16] recently demonstrated that the probability of receiving multimodal therapy (surgery with adjuvant treatment) decreases with age. However, patients in all age groups (40–80+ years) receiving multimodal therapy have an improved 5-year survival compared to patients receiving single modality therapy. Finally, elective neck dissection for oral cavity squamous cell carcinoma (OCSCC) – a technique associated with aesthetic and functional morbidity – also appears to be performed less frequently among older patients [17].

The impact of these therapeutic adaptations made in the real world when managing older patients with OCSCC remains poorly understood. Therefore, we aimed to determine here if recommended treatment (in line with official French practice

guidelines) versus deviation from recommended treatment (including subjective decisions on age and frailty) was beneficial for patients with OCSCC aged >70 years in our center. To do this, we examined how deviation from recommended treatment affected 3-month postoperative morbidity, as well as recurrence and mortality within 5 years.

## Materials and methods

### Study design

A single-center, retrospective, cohort study was conducted on patients aged >70 years and managed for the treatment of p16-negative OCSCC between January 2007 and December 2017 in our tertiary hospital center. Patients included for study were divided into two groups according to treatment type. Treatment choice for all patients was decided by the medical team during multidisciplinary tumor boards.

### Ethics

All procedures performed were part of routine care, and both in accordance with institutional guidelines and with the tenets of the 1964 Helsinki Declaration. Ethical approval was waived by the IRB of Toulouse University Hospital (study reference: RnIPH 2024−10) given the retrospective and non-interventional nature of the study as asserted by the French Jardé Ethical and Regulatory Law. All Participants or their relatives received clear written information and gave free and written informed consent to participate. All data have been anonymized for publication. The study additionally complies with French MR-004 methodology (CNIL 2206723 v 0) covering patient data protection.

### Patient and treatment characteristics

Exclusion criteria were patients with p16-positive OCSCC and with squamous cell carcinoma of the lip. Treatment indications for each patient were developed within multidisciplinary tumor boards, and based on onco-geriatric assessments and national French guidelines. The indication for surgery was based on clinical and radiological data, as well as recommendations from the regional cancer treatment guidelines put forward in 2008 [18]. According to the latter, curative treatment consisted of surgical excision ± lymph node removal (if lymph node metastases were clinically apparent or detected on imaging, or stage cT > 2). After surgery, patient cases were discussed again in multidisciplinary tumor boards for consideration of adjuvant radiotherapy according to TNM classification (>pT2, presence of adenopathy) and histological criteria associated with poor prognosis (<1mm margins, lymphatic emboli, perineural infiltration). Concomitant chemotherapy was indicated for cases with <1mm margins or capsular rupture of lymph nodes. Overall, the treatment was considered recommended when it complied with these aforementioned French guidelines, and deviation from recommended treatment was when it differed from these guidelines. For instance, patients not undergoing the recommended neck dissection or not receiving the recommended adjuvant treatment following subjective decisions made by the multidisciplinary tumor board.

### Data collected

The reasons for deviation from recommended treatment were recorded. Data on other confounding variables were also collected: World Health Organization (WHO) performance status score, age, sex, and type of reconstruction performed. The American Joint Committee on Cancer (AJCC) stage classification was used to classify tumor stages into early (stage I/II) or advanced (stage III/IV) [19].

### Outcome measures

Outcome measures (dependent variables) were 3-month postoperative morbidity, and recurrence and mortality within 5 years after surgery.

 

**Postoperative morbidity.** We defined patients developing Clavien-Dindo grade ≥III complications [20] within 3 months after surgery as patients developing postoperative morbidity. The Clavien-Dindo classification divides postoperative complications from grade I to V according to the need for and type of treatment required, with major complications defined by grade ≥III. Table 1 depicts the different grades and summarizes the clinical situations relevant to our study [21] Grade III complications were pooled for analyses with only the highest-grade complication being used to classify patients. Postoperative tracheostomy was only taken into account when performed to correct grade III medical complications (respiratory distress).

**Recurrence and mortality.** Recurrence was defined by histological confirmation of a local, lymph node, or metastatic recurrence. Recurrence-free survival refers to the time delay between surgery until recurrence or death from any cause. Patients without recurrence or death were censored at the last clinical evaluation. Overall survival was defined as the time between surgery until death, and patients still alive were censored at the last date known alive. Patients lost to follow-up were censored.

## Statistical analyses

Categorical variables are described as frequencies and percentages. We studied the effect of treatment type received (recommended versus deviation from recommended), age (continuous), sex (female versus male), AJCC tumor stage (I–II versus III–IV), WHO performance status (<2 versus ≥2), and the type of reconstruction performed (local flap versus free flap) on the risk of developing postoperative morbidity using multivariate logistic regression. The first model included all variables associated with ≥grade III morbidity in the bivariate analysis with a conservative p-value of 0.20. A backward stepwise selection was performed to obtain the best reduced model. Performance of the logistic model was evaluated using the Hosmer-Lemeshow test.

Recurrence-free survival and overall survival probabilities were estimated using the Kaplan-Meier method and compared using the Log-Rank test. Event-free patients were censored at 5 years of follow-up, and patients lost to follow-up were censored and analysed. A multivariate Cox Proportional Hazard Regression analysis was performed to study disease recurrence and mortality using the same modelling strategy as used for logistic regression. No significant multi-collinearity was detected among the covariates and the proportional hazards assumption was tested a posteriori for each variable included in the Cox models using Schonenfel residual analysis. For this, age was tested as both a categorical and continuous variable.

**Table 1. Definition of Clavien-Dindo postoperative complication grades [20] with examples of complications relevant to our study [21].**

| | Grade | Example clinical situations |
|---|---|---|
| I | Minor complications treated with antiemetics, antipyretic analgesics or intravenous rehydration | • Local treatment of wound disunion<br>• Confusion |
| II | Need of pharmacological treatment other than those authorized above | • Blood transfusion<br>• Phlebitis and anticoagulation treatment<br>• Sepsis and antibiotics |
| III | Complications requiring surgical, endoscopic or radiological treatment | • Gastrostomy<br>• Re-operation (hematoma, abscess, wound debridement) |
| IV | Life-threatening complications | • Emergency tracheotomy<br>• Organ failure<br>• Hemorrhagic shock |
| V | Death | |

Statistical analyses were performed using the GraphPad Prism software version 10.02 (Boston, Massachusetts USA) and the R software (version 4.5.1). P-values <0.05 were considered statistically significant and all tests were two-sided.

## Results

### Demographic and clinical characteristics

A total of 213 patients aged >70 years (79.6 ± 5.7 years) were managed for OCSCC in our center. Among these, n = 28/213(13%) patients were excluded: 1 patient with p16-positive OCSCC and 27 patients indicated palliative treatment. Surgical treatment was therefore indicated for 185 patients according to the official French guidelines. Following subjective decisions made in preoperative multidisciplinary tumor boards, n = 19/185(10%) patients did not undergo neck dissection as initially recommended. Instead, treatment was adapted by considering the age and general frailty of patients. The surgery was performed as recommended among the remaining n = 166/185(90%) patients. Adjuvant treatment with radiotherapy (±chemotherapy) was: 1) recommended for and received by n = 104/166(63%) patients, but 2) not recommended and not received by n = 43/166(26%) patients; thus in line with the guidelines. Adjuvant treatment was recommended but not received by n = 19/166(11%) patients; thus not in line with guidelines. Overall, n = 147/185(79%) patients were included in the recommended treatment group and n = 38/185(21%) patients in the deviation from recommended treatment group. Please refer to Fig 1 for the study flowchart. A full database containing anonymized patient demographic and clinical characteristics can be found in S1 Appendix.

Data on outcomes and independent variables for all included patients (n = 185) are in Table 2. Overall, postoperative morbidity (Clavien-Dindo ≥grade III postoperative complications) had developed by 3 months among n = 51/185(28%) patients. Histologically proven recurrence had developed among n = 101/185(55%) patients (84 censored patients, including 46 lost to follow-up) and n = 88/185(48%) patients (97 censored patients, including 39 lost to follow-up) had died within 5 years after surgery. According to treatment type, post-operative morbidity developed within 3 months among 23% patients in the recommended treatment group and 45% patients in the deviation from recommended group. Recurrence

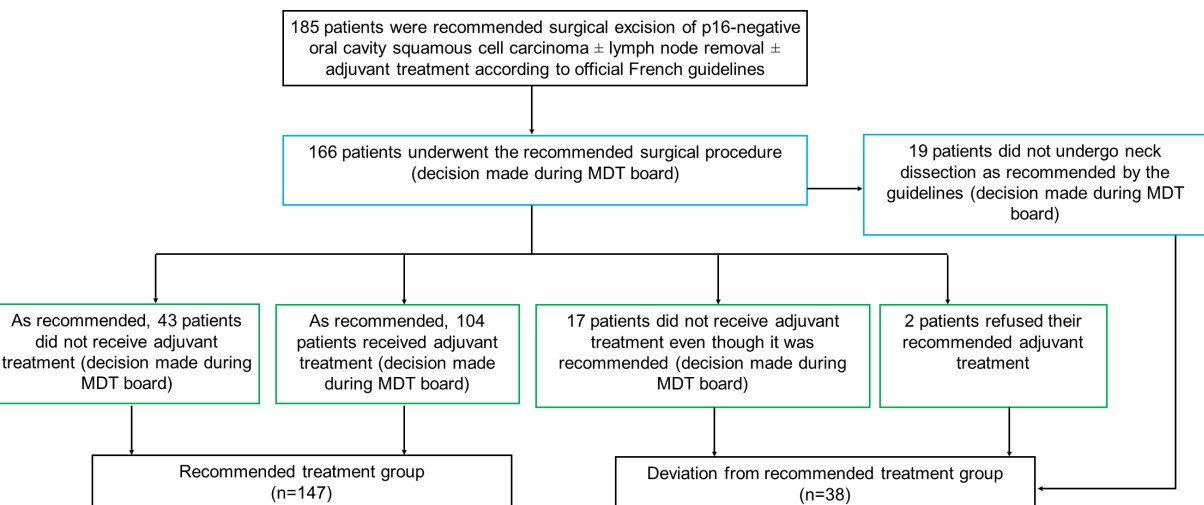

**Fig 1. Study flowchart.** Blue boxes denote preoperative multidisciplinary tumor (MDT) boards and green boxes denote postoperative MDT boards when treatment decisions were made. The indication for surgery was based on clinical and radiological data, as well as official national French recommendations [18]. Curative treatment consisted in surgical excision ± lymph node removal (if lymph node metastases were clinically apparent or detected on imaging, or stage cT > 2). Adjuvant treatment was recommended according to postoperative TNM classification (>pT2, presence of adenopathy) and histological criteria associated with poor prognosis (<1mm margins, lymphatic emboli, perineural infiltration).

**Table 2. The distribution of the independent variables and outcomes (dependent variables) for all treated patients as well as according to recommended treatment and deviation from recommended treatment groups.**

|  | Treatment type | | |
|---|---|---|---|
|  | All | Recommended | Deviation from recommended |
| Patients, n (%) | 185 | 147 (79%) | 38 (21%) |
| **Independent variables** | | | |
| **Age, years** | | | |
| Mean±SD (min, max) | 79.6 ± 5.0 (70, 92) | 78.7 ± 4.7 (70, 90) | 82.7 ± 4.6 (71, 92) |
| **Male sex**, n (%) per group | 73 (39%) | 63 (43%) | 10 (26%) |
| **WHO performance status score,** n (%) per group | | | |
| <2 | 155 (84%) | 129 (88%) | 26 (68%) |
| ≥2 | 30 (16%) | 18 (12%) | 12 (32%) |
| **AJCC tumor stage classification,** n (%) per group | | | |
| I-II | 67 (36%) | 58 (39%) | 9 (24%) |
| III-IV | 118 (64%) | 89 (61%) | 29 (76%) |
| **Reconstruction type performed,** n (%) per group | | | |
| Local flap | 148 (80%) | 116 (79%) | 32 (84%) |
| Free flap | 37 (20%) | 31 (21%) | 6 (16%) |
| **Outcome measures, n (%) per group** | | | |
| 3-month postoperative morbidity | 51 (28%) | 34 (23%) | 17 (45%) |
| Recurrence within 5 years | 101 (55%) | 72 (49%) | 29 (76%) |
| Death within 5 years | 88 (48%) | 62 (42%) | 26 (68%) |

AJCC: The American Joint Committee on Cancer; WHO: World Health Organization.

within 5 years developed among 49% patients in the recommended treatment group and 76% patients in the deviation from recommended group, and death within 5 years occurred among 42% patients in the recommended treatment group and 68% in the deviation from recommended group.

Regarding postoperative morbidity (Table 3), 31 grade III complications developed among n = 27/51(53%) patients developing major complications (grades III–V combined). Note that n = 4 patients developed ≥2 grade III complications. Main grade III complications were wound-healing related (n = 13/27(48%) patients). Among patients developing major complications, n = 16/51(31%) patients developed a grade IV complication and n = 8/51(16%) patients died within 3 months after surgery. All revision surgeries for insufficient resection margins were performed in the deviation from recommended treatment group (n = 4).

### Identification of potential predictive factors

In bivariate analyses (Table 4), a more advanced tumor stage (OR [95%CI]: 5.63 [2.52–14.5]; p < 0.0001) was found significantly positively associated with the development of postoperative morbidity within 3 months after surgery, as well as deviation from recommended treatment (OR [95%CI]: 2.74 [1.34–5.83]; p = 0.006) and free-flap reconstruction (OR [95%CI]: 2.82 [1.33–6.00]; p = 0.006). Considering the results from bivariate analyses stratified by each individual WHO performance status score (2–4; S1 Table) and AJCC tumor stage (S2 Table), as well as the clinical implications [22–24], we binarized these two variables throughout all our regression models.

Table 3. Patients developing postoperative morbidity (Clavien-Dindo complications ≥ grade III) within 3 months after surgery for all treated patients as well as according to recommended treatment and deviation from recommended treatment groups.

| Patients, n (%) | Treatment type | | |
|---|---|---|---|
| | All | Recommended | Deviation from recommended |
| | 185 | 147 (79%) | 38 (21%) |
| **Major postoperative complications, n (%)** | 51 (28%) | 34 (23%) | 17 (45%) |
| **III Complications requiring surgical, endoscopic or radiological treatment, n (%) group** | 27 (53%) | 20 (59%) | 7 (41%) |
| Gastrostomy | 6 | 3 | 3 |
| Drainage of deep fluid collections (abscess, hematoma) | 7 | 1 | 6 |
| Wound healing problems requiring re-operation | 13 | 4 | 9 |
| Revision surgery for insufficient resection margins | 4 | 0 | 4 |
| Revision surgery of the anastomosis (with or without free-flap failure) | 1 | 0 | 1 |
| **IV Life-threatening complication, n (%) group** | 16 (31%) | 9 (26%) | 7 (41%) |
| **V Death, n (%) group** | 8 (16%) | 5 (15%) | 3 (18%) |

Bivariate analyses (Table 4) also demonstrated that deviation from recommended treatment (HR [95%CI]: 2.10 [1.37–3.22]; p = 0.0007) and a more advanced tumor stage (HR [95%CI]: 1.69 [1.12–2.58]; p = 0.01) were found significantly positively associated with 5-year recurrence. Age showed a non-significant positive association with recurrence (HR [95%CI]: 1.03 [1.00–1.10]; p = 0.09). Finally, a more advanced tumor stage (HR [95%CI]: 2.52 [1.53–4.00]; p = 0.0003) and deviation from recommended treatment (HR [95%CI]: 1.85 [1.22–2.34]; p = 0.009) were found positively associated with mortality within 5 years after surgery. Indeed, patients in the deviation from recommended treatment group had approximately a 60% higher risk of mortality within 5 years compared to patients in the recommended treatment group.

## Deviation from recommended treatment is independently associated with increased 3-month postoperative morbidity and recurrence within 5 years

In the multivariate analysis (Table 4), deviation from recommended treatment was found positively associated with major postoperative complications at 3 months (OR$_{adj}$ [95%CI]: 2.63 [1.23–5.82]; p = 0.02). Advanced AJCC tumor stage (OR$_{adj}$ [95%CI]: 4.44 [1.92–11.6]; p = 0.001) and free-flap reconstruction (OR$_{adj}$ [95%CI]: 2.54 [1.10–5.70]; p = 0.02) were also found positively associated with the development of major postoperative complications. The event:variable ratio was 17 for this model.

Proven recurrence developed among n = 72/147 (49%) patients in the recommended treatment group and among n = 29/38 (76%) patients in the deviation from recommended treatment group (Table 2). Recommended treatment resulted in a significantly higher recurrence-free survival rate (46.0% [95%CI 38.0–55.6]) compared to deviation from recommended treatment (17.2% [95%CI 8.2–36.2]) (Log-Rank p = 0.00052) (Fig 2a), reflecting a lower cumulative recurrence rate of 56.0% versus 82.8%.

Cox Proportional Hazard Regression analysis for the effects of the independent variables on the risk of recurrence (Table 4) demonstrated that deviation from recommended treatment was independently associated (HR$_{adj}$ 1.79 [1.14–2.83]; p = 0.01) with significantly higher risk of recurrence within 5 years. More advanced AJCC tumor stage (HR$_{adj}$ 1.57 [1.02–2.42]; p = 0.04), but not age (HR$_{adj}$ 1.04 [1.92–1.05]; p = 0.29), was also independently associated with higher risk of recurrence within 5 years. The proportional hazards assumptions were met for all variables in this 5-year recurrence model with age as a continuous variable (global test p = 0.81) (S1a Fig). The event:variable ratio was 33.7 for this model and the model significantly improved fit (likelihood ratio test p = 0.001 and concordance 0.618). Finally, an exploratory multivariate model, not adjusted for age or WHO status score, further confirmed that deviation from recommended treatment was significantly associated with increased risk of recurrence within 5 years (S3 Table).

**Table 4. Bivariate and multivariate regression analysis results demonstrating the predictors of postoperative morbidity within 3 months (Clavien-Dindo complications ≥ grade III), recurrence or mortality within 5 years after surgery.**

| Independent variables | 3-month postoperative morbidity | | | |
|---|---|---|---|---|
| | Bivariate | | Multivariate | |
| | OR [95%CI] | p value | OR$_{adj}$ [95%CI] | p value |
| Deviation from recommended treatment | 2.74 [1.34-5.83] | 0.006* | 2.63 [1.23-5.82] | 0.02 |
| Age | 0.97 [0.91-1.03] | 0.29 | – | – |
| Sex (male) | 0.91 [0.53-1.8] | 0.91 | – | – |
| WHO performance status score ≥2 | 1.24 [0.53-2.81] | 0.58 | – | – |
| AJCC tumor classification III-IV | 5.63 [2.52-14.5] | <0.0001* | 4.44 [1.92-11.6] | 0.001 |
| Free-flap reconstruction | 2.82 [1.33-6.00] | 0.006* | 2.54 [1.10-5.70] | 0.02 |
| **5-year recurrence** | | | | |
| Independent variables | Bivariate | | Multivariate | |
| | HR [95%CI] | p value | HR$_{adj}$ [95%CI] | p value |
| Deviation from recommended treatment | 2.10 [1.37-3.22] | 0.0007* | 1.79 [1.14-2.83] | 0.01 |
| Age | 1.03 [1.00-1.10] | 0.09* | 1.04 [0.92-1.05] | 0.29 |
| Sex (male) | 0.96 [0.71-1.53] | 0.85 | – | – |
| WHO performance status score ≥2 | 1.15 [0.62-1.93] | 0.59 | – | – |
| AJCC tumor classification III-IV | 1.69 [1.12-2.58] | 0.01* | 1.57 [1.02-2.42] | 0.04 |
| Free-flap reconstruction | 1.17 [0.70-1.82] | 0.51 | – | – |
| **5-year mortality** | | | | |
| Independent variables | Bivariate | | Multivariate | |
| | HR [95%CI] | p value | HR$_{adj}$ [95%CI] | p value |
| Deviation from recommended treatment | 1.85 [1.22-2.34] | 0.009* | 1.35 [0.82-2.23] | 0.2 |
| Age | 1.04 [1.00-1.08] | 0.032* | 1.05 [1.00-1.09] | 0.04 |
| Sex (male) | 1.31 [0.9-1.9] | 0.30 | – | – |
| WHO performance status score ≥2 | 1.34 [0.82-2.10] | 0.34 | – | – |
| AJCC tumor classification III-IV | 2.52 [1.53-4.00] | 0.0003* | NA | NA |
| Free-flap reconstruction | 0.75 [0.54-1.21] | 0.20* | 1.35 [0.82-2.25] | 0.3 |

HR: hazard ratio; HR$_{adj}$: adjusted Cox Proportional hazard ratio; NA: not applicable as the model was stratified according to AJCC tumor stage; OR: odds ratio; OR$_{adj}$: adjusted odds ratio.

*: variables with p<0.20 in bivariate analyses and thus included in corresponding multivariate models.

Within 5 years after surgery, n = 62/147(42%) patients had died in the recommended treatment group and n = 26/38(68%) patients in the deviation from recommended treatment group (Table 2). Recommended treatment resulted in a significantly higher overall survival rate (54.0% [95%CI 46.0–63.3]) compared to deviation from recommended treatment (29.3% [95%CI 8.2–36.2]) (Log-Rank p = 0.008) (Fig 2b), reflecting a lower cumulative survival rate of 46.0% versus 70.7%.

A significant violation of the proportional hazards assumption was detected for the AJCC tumor stage variable ($\chi^2$ = 5.22, p = 0.022) in the 5-year mortality model with age as a continuous variable (S1b Fig). The Cox Proportional Hazard Regression mortality model was therefore stratified according to AJCC tumor stage (Table 4), with an event:variable ratio was 22 and the model significantly improved fit (likelihood ratio test p = 0.03 and concordance 0.594). In all, only age was found independently associated (HR$_{adj}$ 1.05 [1.00–1.09]; p = 0.04) with higher risk of mortality within 5 years after surgery. Deviation from recommended treatment was not associated with increased risk of mortality within 5 years (HR$_{adj}$ 1.35 [0.82–2.25]; p = 0.2). Deviation from recommended treatment was only found significantly associated with an increased risk of mortality when age was not adjusted for (S3 Table).

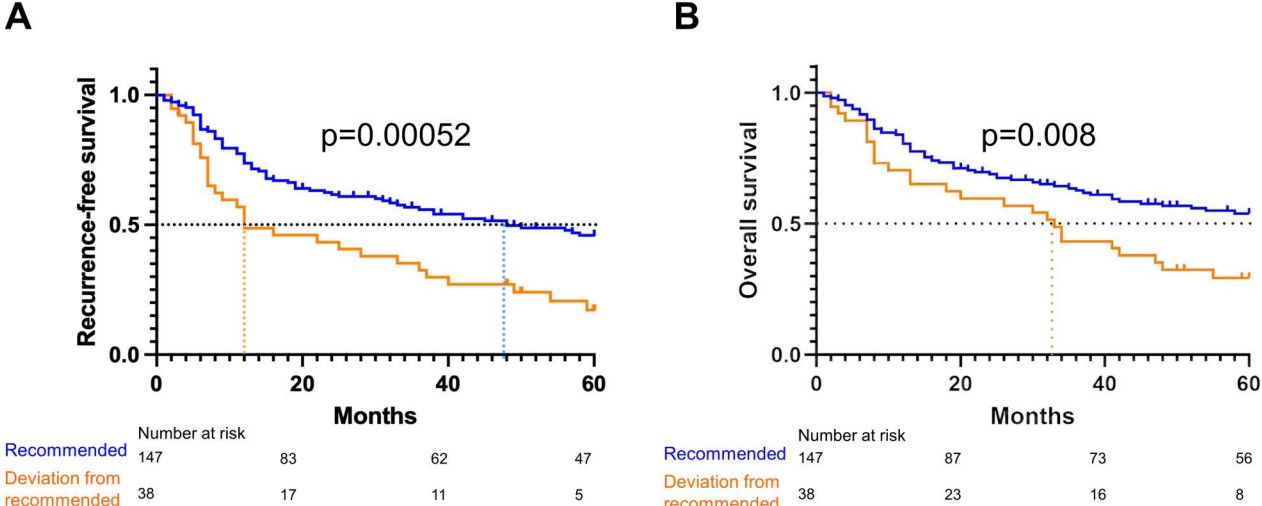

**Fig 2. Kaplan-Meier curves for: 5-year recurrence-free survival (a) and 5-year overall survival (b) according to recommended treatment or deviation from recommended treatment.** The dotted lines indicate median time delays in months for each treatment group.

## Discussion

We have been indicating personalized curative treatment for "frail" patients aged >70 years with OCSCC in our center following subjective decisions made during multidisciplinary tumor boards. In this study, n = 147/185 (79%) patients received the recommended treatment according to official French guidelines. However, a subgroup of patients (n = 38/185;21%) did not receive the recommended neck dissection and/or adjuvant treatment. Hence, they received a treatment that was not in line with official guidelines, but was considered adapted to patient age and frailty. We investigated whether this recommended versus deviation from recommended treatment was beneficial for the treated patients concerned. Overall, patients in the deviation from recommended treatment group had a significantly reduced disease-free survival and overall survival. After controlling for confounding factors, we demonstrate that deviation from recommended treatment was also independently associated with increased development of postoperative morbidity at 3 months and increased risk of recurrence within 5 years after surgery.

Firstly, our findings are in line with similar studies in the literature. For instance, Sanabria et al. [14] including 305 patients aged >70 years with head and neck cancer. The authors found that 19% of patients were under-treated, similar to 21% in our study here. The overall and cancer-specific survival rates of those under-treated were inferior to their control group. However, specific details on how under-treatment varied from clinical guidelines were not given. Dronkers et al. [25] described how 17% of patients with primary HNSCC (total inclusion n = 829) received non-standard curative treatment after decisions taken by the multidisciplinary medical team (11%) or patient choice (6%). This treatment regimen, for which deviations from standard treatment were again not specified, also resulted in reduced 3-year survival.

We show that patients receiving less aggressive treatment were significantly more likely to have developed major postoperative complications within 3 months after surgery. Discrepancies between personal opinions in the medical team and clinical examination results thus highlight a major decision-making bias. We therefore further questioned our practices by performing additional analyses to determine which clinical variables prompted us to indicate non-standard treatment. A multivariate regression analysis for deviation from recommended treatment showed that age and WHO performance status score were decisive variables (S4 Table). Hence, we more often indicated non-standard treatment to patients: (1) with WHO scores ≥2 (OR$_{adj}$ [95%CI]: 1.10 [0.10–1.92]; p = 0.02); and (2) who were older (1.10 [1.02–1.22]; p = 0.01). Indeed,

the reasoning found in corresponding patient medical records was general "tiredness", "old age", and "frailty" (estimated by WHO score). Similarly, Dronkers et al. [25] also put forward arguments related to frailty for favoring non-standard treatment. The authors also identified risk factors for receiving non-standard treatment for curative treatment of HNSCC, with patients living alone, with multiple co-morbidities or advanced tumor stage, females, and older patients being identified as at higher risk of receiving non-standard treatment [25].

We cannot explain at this point why the absence of lymph node removal or adjuvant treatment was independently associated with increased 3-month postoperative morbidity in our cohort. However, we are aware that this finding must be interpreted with caution. Firstly, approximately half of the deviation from recommended treatment group did not receive the recommended adjuvant treatment following decisions made in postoperative multidisciplinary tumor boards. These decisions were thus inevitably taking into account increased frailty, actually influenced by the development of postoperative complications for some patients, and not due to under-treatment. Secondly, patient frailty was not identified in bivariate analyses and was therefore not included in our multivariate models. Interestingly, a WHO score ≥2 was not found positively associated with 3-month postoperative morbidity but with deviation from recommended treatment (S4 Table), highlighting thus the substantial source of bias in our study. Indeed, as mentioned above, patients not receiving the recommended treatment were generally older and in a generally worse overall condition. Only a randomized trial comparing patients receiving and not receiving recommended treatment could reliably deduce differences in risks of developing complications, recurrence, and mortality. In the future, we could also additionally use the more specific G8 questionnaire for general patient frailty [26]. This was only integrated systematically after 2015 in pre-operative onco-geriatric assessments.

The independent association of deviation from recommended treatment with a significantly higher risk of recurrence within 5 years can be partly explained by poorer control of lymph node cancer growth. The absence of lymph node removal in cT > 2 and/or cN+ patients and/or the absence of adjuvant treatment in pT > 2 and/or pN+ patients within this patient group is likely detrimental. To date, there is no consensus on performing lymph node removal in patients with cT1/2 or with cN0, and this is regardless of age [27]. However, our study shows that the benefit/risk balance is not in favor of withholding treatment in older patients with cervical adenopathy and/or tumor size >4 cm.

Other studies point the finger at surgeon negligence toward preoperative geriatric assessment. While functional status is an important predictor of surgical outcome, a formal comprehensive preoperative assessment has not been established and most surgeons do not measure preoperative frailty [28]. The role of the geriatrician and their participation in multidisciplinary meetings have yet to be defined. Indeed, this was highlighted by a survey sent out to HNSCC radiation oncologists in Italy. Results showed that only 6.9% of multidisciplinary meetings regularly include a geriatrician. The authors attributed this to the difficulty of integrating a full geriatric assessment into daily clinical practice [29].

We also performed a complementary multivariate analysis to identify variables associated with the indication of free-flap reconstruction in our study (n = 37) with age as a continuous variable (S5 Table). A more advanced tumor stage ($OR_{adj}$ [95%CI]: 2.70 [1.17–6.88]; p = 0.03) and younger age ($OR_{adj}$ [95%CI]: 0.93 [0.87–1.00]; p = 0.04) were predictive of free-flap reconstruction. A higher WHO performance status score was thus not identified as a predictive factor for this reconstruction type in our team. Free-flap reconstruction surgery appears to be a safe technique among older patients, with surgical complications comparable to those in younger patient populations. However, due to concomitant medical conditions, postoperative complications are more frequent in older patient groups, thus resulting in longer hospital stays [30,31] and undoubtedly reduced indications among older patients despite a lack of official contraindications. Some authors recommend multimodal preoperative education programs to increase functional capacity and potentially reduce postoperative complications [32].

Free-flap reconstruction was found independently associated with increased 3-month postoperative morbidity within 3 months ($OR_{adj}$ 2.54 [1.10–5.70], p = 0.02). Nonetheless, we believe that free-flap reconstruction allows for higher-quality resection with wider margins than when a local flap is used. We also believe that free-flap reconstruction allows for a more functional reconstruction, which would also affect overall survival. However, surgery in older patients most often consists

of oncologic resection without functional reconstruction [33]. Restoration of function is as important as tumor control, and even more so in older patients. Akishita et al. [3] surveyed healthcare providers caring for older patients and their patients/families in Japan on expected priorities from treatment. The results showed that patients preferred the maintenance of autonomy and preservation of function, while the medical team favored curing.

Finally, a national Australian study has recently determined the characteristics of clinicians most often complying with clinical practice guidelines in oncology. Interestingly, younger clinicians and those attending several multidisciplinary tumor boards per week were more frequently using official recommendations [34].

## Conclusions

In all, we can see that the subjective decisions taken in multidisciplinary tumor boards do not always agree with official recommendations, and the reasons for these discrepancies cannot always be justified. Being aware of our past subjective decisions, and especially their consequences, will guide our future decision-making towards more objective decisions based on medical assessments. We wish to emphasize the importance of validating preoperative risk stratification models in older patients to better define appropriate and consistent treatment regimens.

## Supporting information

**S1 Appendix. Full database of anonymized patient relevant demographic and clinical characteristics.**
(XLSX)

**S1 Fig. Schoenfeld residual plots.** The proportional hazard assumptions were tested for the recurrence (a) and mortality (b) models with age as a continuous variable.
(TIF)

**S1 Table. Bivariate analyses for 3-month postoperative morbidity.** Modelled according to each individual WHO performance status score.
(DOCX)

**S2 Table. Bivariate analyses for 3-month postoperative morbidity.** Modelled according to each individual AJCC tumor classification grade.
(DOCX)

**S3 Table. Exploratory multivariate Cox models not adjusted for age or WHO performance status score.**
(DOCX)

**S4 Table. Multivariate model for deviation from recommended treatment.**
(DOCX)

**S5 Table. Multivariate model for free-flap reconstruction.**
(DOCX)

## Acknowledgments

We thank Dr Valérie Lauwers-Cances for assistance with statistical analyses.

## Author contributions

**Conceptualization:** Alice Prevost, Frédéric Lauwers, Franck Delanoë.

**Data curation:** Alice Prevost, Hugo Poncet, Victor Benvegnu, Vinciane Poulet.

**Formal analysis:** Alice Prevost, Hugo Poncet, Jacqueline Butterworth, Andréa Varazzani, Frédéric Lauwers, Franck Delanoë.

**Methodology:** Alice Prevost, Franck Delanoë.

**Supervision:** Franck Delanoë.

**Writing – original draft:** Alice Prevost.

**Writing – review & editing:** Alice Prevost, Hugo Poncet, Victor Benvegnu, Vinciane Poulet, Jacqueline Butterworth, Andréa Varazzani, Frédéric Lauwers, Franck Delanoë.

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
