## [Decision Letter · Decision Letter 0]

26 May 2025

Dear Dr. Prevost,

Thank you for submitting your manuscript to PLOS ONE. After careful consideration, we feel that it has merit but does not fully meet PLOS ONE’s publication criteria as it currently stands. Therefore, we invite you to submit a revised version of the manuscript that addresses the points raised during the review process.

We look forward to receiving your revised manuscript.

Kind regards,

John Minh Le, MD, DDS

Academic Editor

PLOS ONE

Journal Requirements:

2. In the online submission form, you indicated that [The original patient data and datasets generated and analyzed in this study are available from the corresponding author on reasonable request and after IRB approval of data transfer.].

4, Please review your reference list to ensure that it is complete and correct. If you have cited papers that have been retracted, please include the rationale for doing so in the manuscript text, or remove these references and replace them with relevant current references. Any changes to the reference list should be mentioned in the rebuttal letter that accompanies your revised manuscript. If you need to cite a retracted article, indicate the article’s retracted status in the References list and also include a citation and full reference for the retraction notice.

Reviewers' comments:

Reviewer's Responses to Questions

**Comments to the Author**

1. Is the manuscript technically sound, and do the data support the conclusions?

Reviewer #1: Partly

Reviewer #2: Partly

2. Has the statistical analysis been performed appropriately and rigorously?

Reviewer #1: Yes

Reviewer #2: No

3. Have the authors made all data underlying the findings in their manuscript fully available?

Reviewer #1: No

Reviewer #2: Yes

4. Is the manuscript presented in an intelligible fashion and written in standard English?

Reviewer #1: Yes

Reviewer #2: Yes

Reviewer #1: Dear authors,

I have gladly reviewed your manuscript on this important topic. Please find my comments attached:

Kind Regards,

Ubai Alsharif

General issues for the whole manuscript:

- Please point that the official guidelines are the French national guidelines in the abstract and in the methods as well.

- I think the terms "standard" and "non-standard" treatments should be replaced with another term such as "MDTB recommended treatment" or "Adherence to treatment recommendation" vs. "non adherence". Also, I don' think the term need to be abbreviated.

- please replace the term "lymph node removal" with "neck dissection" in the manuscript and in the figures.

- The word "preoperative" is written as one word.

- I am not aware of a perioperative neck dissection. Neck dissection is usually done in the same operation with tumor resection.

- The manuscript is well written. However, there are minor grammatical errors and the flow of the text can be improved. If possible, it would help to revise the text once.

Abstract:

- please rephrase the sentence "we found surgical ..." to "We included 185 patients who underwent surgical resection of OCSCC".

- Cox regression is not used to calculate the 5-year mortality or recurrence risk. The probability of death is estimated by using Kaplan-meier Method for overall survival. For recurrence, cumulative incidence function should be used. Cox regression is used to estimate hazards ratios of certain variables.

- The sentence "with a tendency for a higher risk of 5-years mortality" is wrong. In a cox regression, the HR reflects an association between a variables S/NS with an outcome "Mortality" after adjustment. It has nothing to do with duration "5 years", and there are no tendencies with such a wide CI.

Introduction:

- Line 75: A doctor is a person with doctoral degree. The authors mean "Medical practitioner or physician".

Methods:

- I urge the authors to redo the statistical analysis while stratifying WHO performance status score by point 1,2 and more than two, and AJCC tumor stage into I, II, III, and IV. This would allow for a better comparison of results for future studies.

- The adherence to treatment is probably associated with age. We should question whether it's correct to include age and treatment as variables in the cox regression. The proportional hazards assumption needs to be tested for age as categorical variables, and it would not hold probably. If this is the case, then the cox regression has to be stratified by age as well. Only this would show the true HR for treatment especially in the overall survival model. I urge the authors to re-run the statistical analysis after considering my recommendation in the results section as well.

- The authors should keep in mind that there is confounding by indication in this study. Patients who did not adhere to a certain therapy, probably did not adhere in the first place because of a worse general health condition or a higher age. The adherence in itself is a result of age and general condition. So is incorporating age and general condition into the statistical models correct in the beginning? This might be worth to incorporate as an additional model in the appendix.

Results:

- Please state why the 19 patients did not undergo neck dissection. Did they refuse or was it the surgeon?

- There is a redundancy in the paragraph of the lines 187-194. I think it can be simplified with a referral to figure 1.

- Table 2: Please split Table 2 into 3 columns NS group, S group and overall, and report all variables stratified by treatment. In addition, please put the outcome measures after the independent variables.

- In possible, please list the WHO performance status score by point 1,2 and more than two, and AJCC tumor stage into I, II, III, and IV.

- if available report the histological criteria associated with poor prognosis in your group in Table 2.

- Table 3: Please stratify here also by treatment group. I would add the outcome measures from Table 2 to Table 3.

- The sentence "Indeed, .." in lines 246-248 is wrong. This is not the interpretation of HR of 1.6. Please correct.

- There is no need for the new titles on the liens 250-251 and 258-259 and 273-274. Consider using one subtitle for all there paragraphs.

- There is no need to report two digits after the comma in the CI when it's higher than 10 as in Line 262.

Discussion:

- The sentence in the lines 298-299 is not true. There is no tendency.

- the statement in the lines 329 - 331. Well as I mentioned before, there is a reason why the patients did not receive the full treatment and it's probably the reason fo the postoperative morbidity. It is probably the general condition.

Figure 1:

I would suggest improving the figure a bit, by adding two blocks that show preoperative and postoperative MDTB. In addition, I suggest using the terms adhered to the postoperative MDTB recommendation.

Reviewer #2: I believe the argument for the study could be made more forcefully. For example: "Although clinicians often deviate from standard treatment guidelines when managing older patients with OCSCC—relying instead on subjective judgments of age and frailty—the impact of these real-world, non-standard decisions on patient outcomes remains poorly understood."

In the methods section, the authors need a clear account of how they handled missing data. This is critical in retrospective studies. Also, the methods section and the results section could use a little reorganization. In the methods section, the authors should clearly state the number of outcome events expected, justify the number of predictors retained per model, and acknowledge the risk of overfitting, especially for the morbidity model. In the discussion, the authors should reflect on the model power, especially the weak mortality result (HR 1.3, CI includes 1, p=0.31).

Furthermore, the authors MUST address the assumptions for the inferential statistics. Were all the assumptions met for the Cox proportional hazard and logistic regression? If all assumptions were met, a global statement would suffice. If not, then the authors should explain why they proceeded with the study.

In the results, there is no discussion of statistical power, nor is there a post-hoc power analysis. So, it is not clear if the study had sufficient power to detect the observed effects. Lastly, the risk of overfitting in the morbidity model is not adequately discussed, considering the relatively small number of events. This could undermine the reliability of the model’s findings.

Overall, I found the manuscript well-written with only minor stylistic errors. I hope to see a revised draft soon.

**Do you want your identity to be public for this peer review?** For information about this choice, including consent withdrawal, please see our Privacy Policy

Reviewer #1: **Yes: ** Ubai Alsharif

Reviewer #2: No

---

## [Author Response · Author response to Decision Letter 1]

10 Jul 2025

Please see the pdf version attached to our resubmission, with tables and figures imbedded.

---

## [Decision Letter · Decision Letter 1]

31 Jul 2025

Curative treatment incorporating subjective decisions on age and frailty is not beneficial for older patients with oral cavity squamous cell carcinoma

PONE-D-25-01028R1

Dear Dr. Prevost,

We’re pleased to inform you that your manuscript has been judged scientifically suitable for publication and will be formally accepted for publication once it meets all outstanding technical requirements.

Kind regards,

John Minh Le, MD, DDS

Academic Editor

PLOS ONE

Additional Editor Comments (optional):

Thank you for your patience and time in making the revisions for this manuscript. The reviewers have placed their final comments below and believe that it is suitable for acceptance. Congratulations. 

Reviewers' comments:

Reviewer's Responses to Questions

**Comments to the Author**

Reviewer #1: All comments have been addressed

Reviewer #2: All comments have been addressed

2. Is the manuscript technically sound, and do the data support the conclusions?

Reviewer #1: Yes

Reviewer #2: Yes

3. Has the statistical analysis been performed appropriately and rigorously?

Reviewer #1: Yes

Reviewer #2: Yes

4. Have the authors made all data underlying the findings in their manuscript fully available?

Reviewer #1: Yes

Reviewer #2: Yes

5. Is the manuscript presented in an intelligible fashion and written in standard English?

Reviewer #1: Yes

Reviewer #2: Yes

Reviewer #1: Dear authors,

The manuscript shows substantial improvement after revision—well done and congratulations. I recommend accepting it.

If possible, I would suggest shortening the subtitle on lines 293–295, as it reads more like a sentence and a result than a proper subtitle.

Best wishes

UA

Reviewer #2: The authors addressed my questions regarding the logistic regression assumptions and the statistical power, but I would like to touch on what appears to be a sensitive topic. I understand that the event-per-variable rule is a useful guideline for model stability. However, I strongly encourage the authors to consider a brief post-hoc power analysis using G*Power in their future endeavors. I feel that this would be helpful in transparency as well as reduce the number of questions during a review. Programs like G*Power are free, quick and easy to use, and can be used for retrospective studies.

I thank the authors for the time they spent on addressing my questions. I wish them well.

**Do you want your identity to be public for this peer review?** For information about this choice, including consent withdrawal, please see our Privacy Policy

Reviewer #1: **Yes: ** Ubai Alsharif

Reviewer #2: No

---

## [Editor Report · Acceptance letter]

PONE-D-25-01028R1

PLOS ONE

Dear Dr. Prevost,

I'm pleased to inform you that your manuscript has been deemed suitable for publication in PLOS ONE. Congratulations! Your manuscript is now being handed over to our production team.

Kind regards,

on behalf of

Dr. John Minh Le

Academic Editor

PLOS ONE